# Base Flow Variation and Attribution Analysis Based on the Budyko Theory in the Weihe River Basin

Zheng Mu [1,†], Guanpeng Liu [1,†], Shuai Lin [1], Jingjing Fan [1,2,*], Tianling Qin [2,*], Yunyun Li [3], Yao Cheng [1] and Bin Zhou [1]

1. Hebei Collaborative Innovation Center for the Regulation and Comprehensive Management of Water Resources and Water Environment, Hebei University of Engineering, Handan 056002, China; muzheng1981@hebeu.edu.cn (Z.M.); liugp1998@163.com (G.L.); as18833022875@163.com (S.L.); chengyao@hebeu.edu.cn (Y.C.); zhoubin@meichao.com (B.Z.)
2. State Key Laboratory of Simulation and Regulation of Water Cycle in River Basin, China Institute of Water Resources and Hydropower Research, Beijing 100038, China
3. College of Resources and Environmental Engineering, Mianyang Teachers' College, Mianyang 621000, China; liyunyun19900627@163.com
* Correspondence: fanjingjing@hebeu.edu.cn (J.F.); qintl@iwhr.com (T.Q.)
† These authors contributed equally to this work.

**Abstract:** The composition and change of runoff are closely related to climate change and human activities. To design effective watershed water resources management measures, there is a need for a clear understanding of the impact of climate change and human activities on baseflow and surface runoff. The purpose of this essay is to quantify their impact on the annual total stream flow, surface runoff, and base flow in the Weihe River Basin (WRB) using a two-stage annual precipitation partitioning method, wherein the surface runoff and base flow are separated from the measured total flow by using a one-parameter digital filter method for which the common filter parameter value is 0.925. The stream flow records were split into two periods: 1960–1970 (pre-change period) and 1971–2005 (post-change period) based on the hydrological breakpoints detected. We found that climate change and human activities have different impacts on base flow and surface runoff. We attributed the decrease in surface runoff due to climate change accounting for 76–78%, while we determined that human activities were responsible to the decrease in base flow accounting for 59–73% of the total observed change. We concluded that both climate change and human beings contributed to the hydrologic change through different hydrological processes: climate change dominated the surface runoff change, while human influences controlled the base flow change. To achieve the expected goals of ecological restoration, appropriate measures must be taken by watershed management in the WRB to mitigate the likely impacts of climate change on water hydrology.

**Keywords:** climate change; human activities; base flow; surface runoff; Weihe River Basin

## 1. Introduction

The global hydrologic cycle and distribution of water resources are changing on various scales due to climate change and human interference during the past decades [1–3]. The changes in precipitation, temperature, wind speed, humidity, and solar radiation have caused decreases in streamflow in some regions [4,5]. Land use changes to the vegetation distribution structure lead to changes in evapotranspiration and, thus, lead to changes in runoff [6,7]. Dam construction will affect the flow of the river [8,9]; dam and reservoir construction will reduce runoff [10,11]. Deforestation will lead to a decrease in interception capacity and an increase in runoff, while afforestation will lead to an increase in runoff infiltration, a decrease in runoff, and an increase in underground runoff [12,13]; Agricultural water management leads to reduced runoff or groundwater [14]. Therefore, under the influence of the above factors, there is a variation in runoff in different parts of

the world. Quantifying the individual impacts of climate change and humans is important for mitigating the negative effects and adapting to novel environments in the future.

Numerous studies have been conducted to quantify the individual effects of climate change and human activities on water resources in different parts of the world [15–19] including the Tarim River [20], Yangtze River [21], and Shiyang River in China [22], the Nzoia River in Kenya [23], and the Athabasca River in Canada [24]. The study in the Haihe River Basin showed that the runoff in the Taolinkou, Zhangjiafen, and Guantai basins decreased by 41.5%, 59.9%, and 73.9%, respectively, mainly affected by human beings [25]. Ma et al. [26] and Zheng et al. [27] reported that climate change contributed over 50% of the decrease in the inflow to the Miyun reservoir during the past decade. Using data from 413 watersheds in the USA, Wang and Hejazi [28] showed that the impacts of climate change outweighed the effects of human activities. Analysis in the Cimarron Skeleton watershed (the South-central Great Plains) in the USA indicated that changes in water use, land use, and land cover accounted for approximately 50% of changes in water flows. [29]. More recently, the study in the Luan River Basin in China illustrated that the impact of climate change contributed 44% in the wet season, but human activities contributed 93% in the dry season to the flow change [30].

The Weihe River Basin (WRB), an important river in Shaanxi Province, known to the Shaanxi people as their mother river, is the largest tributary of the Yellow River in northern China and is a main river for Ningxia and Gansu Provinces in northwestern China that provides the surface water for irrigation and water supplies for several major cities in the arid region. The stream flow in the WRB has significantly declined in the 20th century [31,32]. Based on 13 stream flow gauges in the WRB, the monthly stream flow depth has declined 0.1–2.1 mm during the period from 1960 to 2009, and the main driving factors for the declined stream flow are believed to be related to reservoir operation, vegetation change, surface water consumption, and water and soil conservation [33]. Using an improved climate elasticity method, Zhan [34] found that human activities contributed to 71–78% of the stream flow decline. More recently, Jiang [35] used a framework to identify the two effects on stream flow across the WRB and found that human activities had a significant impact on decreasing runoff.

Most previous studies have focused on the total stream flow, and little attention has been given to how climate or humans alter the flow pathways, i.e., surface flow and groundwater. The arid regions are dominated by rainfall-excess hydrology, and the surface flow generation has rather different mechanisms from the base flow, which is derived from groundwater. We presume that both climate change and human activities may contribute differently to the changes in surface runoff and base flow. Quantifying the effects of climate change and human activities on water resources enables addressing and better management of the water crisis faced by arid regions.

The objective of this study was to quantify the contributions of climate change and human activities to the interpretation of surface runoff and baseflow observations at three hydrologic gauging stations in the WRB. The base flow and surface runoff components were estimated using a two-stage annual precipitation partitioning method. The spatial variation patterns of climate change and human activity with respect to their influences on the base flow and surface flow were also discussed. The results provide a comprehensive analysis climate and human effects on base flow and surface runoff in the WRB.

## 2. Materials and Methods

### 2.1. Study Area and Hydrometeorological Data

The WRB (104°00′ E~110°20′ E, 33°50′ N~37°18′ N), located in the semiarid region of China, has a watershed area of 135,000 km$^2$ and a main river length of 818 km (Figure 1). The WRB starts in the Niaoshu Mountain, flows through Gansu, Ningxia, Shanxi, and eventually joins the Yellow River at Tongguan. The average annual natural runoff is 10.4 billion m$^3$. The average annual precipitation is about 610 mm, the average annual tem-

perature ranges from 7.8 to 13.5 °C, and the average annual potential evapotranspiration (PET) varies from 800 to 1200 mm [36].

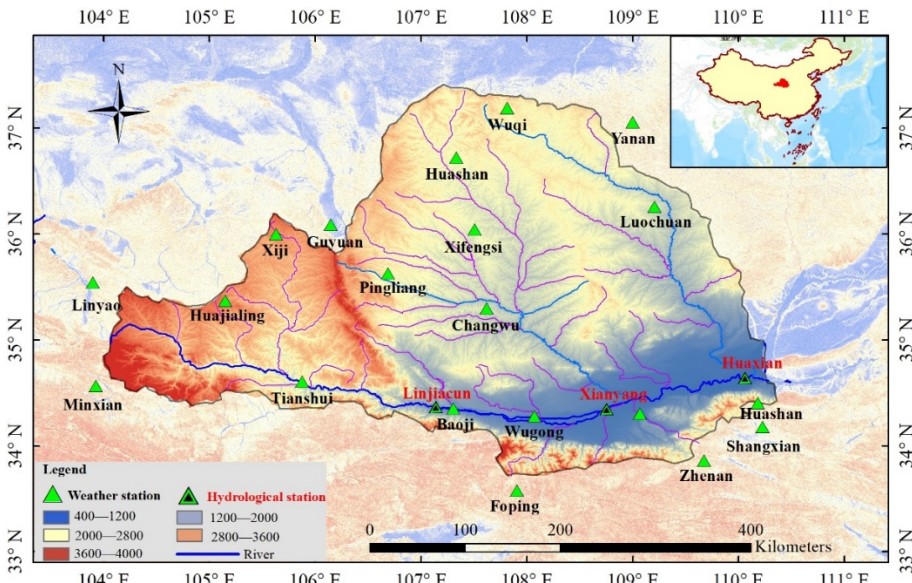

**Figure 1.** The spatial distribution of 3 hydrologic stations and 22 rain gauges in the Weihe River Basin.

In this study, 22 meteorological stations and three hydrological stations were used to derive daily precipitation, temperature, and stream flow data in the WRB. These stations were located throughout the WRB, representing the characteristics of climate and hydrology for each catchment. Daily data on precipitation and other meteorological variables for the period from 1960 to 2005 were provided by the China Meteorological Administration. The monthly PET series were calculated using the Penman-Monteith method [37] for every meteorological station. The areawide precipitation and PET for each hydrological station was generated by averaging the readings across meteorological stations in and around the controlled watershed. Shaanxi hydrology and Water Resources Bureau provided daily flow data of major rivers in PNG from 1960 to 2005.

The one-parameter digital filter decomposed the daily stream flow into surface runoff and baseflow components with the filter parameter value of 0.925 [38]. In this study, the daily PET, rainfall, surface runoff, and base flow were summarized as annual scale values. The actual annual evaporation values were computed as residuals of the water balance, E = P − Q, where E is evapotranspiration, P is precipitation, and Q is streamflow.

### 2.2. Digital Filtering

The digital filtering method was used to segment the daily scale runoff data of Weihe River Basin. Digital filtering is derived from signal processing technology, and its main principle is the combination of runoff division and digital signal analysis. The fast response and slow response signals in the process of precipitation and runoff are decomposed into high frequency signals and low frequency signals by a digital filter, which represent surface runoff and underground runoff, respectively [39].

The Lyne–Hollick filtering method was first proposed by Lyne and Hollick in 1979. Nathan and Mcmahon introduced hydrologic calculations into the method in 1990; its segmentation equation is:

$$Q_{dt} = f_1 Q_{d\ (t-1)} + \frac{1+f_1}{2}\left[Q_t - Q_{(t-1)}\right], \tag{1}$$

$$Q_{bt} = Q_t - Q_{dt}, \tag{2}$$

where $Q_{dt}$ and $Q_{dt-1}$ are the surface runoff at time $t$ and time $t - 1$, respectively; $Q_t$ and $Q_{(t-1)}$ are the runoff at time $t$ and time $t - 1$, respectively; $Q_{bt}$ is the base flow at time $t$; and $f_1$ is the filter parameter value range 0.90 to 0.95; Nathan selected the filter parameter 0.925 by comparison [38].

### 2.3. Mann-Kendall Inspection

The Mann-Kendall trend test is a nonparametric statistical test method, also known as the no distribution test [40,41]. For the time series $X = (X_1, X_2, X_3, \ldots, X_n)$, ($n$ is the number of variables), the Mann-Kendall trend test statistic is S.

$$\text{S} = \sum_{j=1}^{n-1} \sum_{i=j+1}^{n} \text{sgn}\,(X_i - X_j), \tag{3}$$

$$\text{sgn}(X_i - X_j) = \begin{cases} 1, & X_i - X_j > 0 \\ 0, & X_i - X_j = 0 \\ -1, & X_i - X_j < 0 \end{cases} \tag{4}$$

$$\begin{cases} Z = \dfrac{S+1}{\sqrt{\frac{n(n-1)(2n+5)}{18}}}; & S > 0 \\ Z = 0 & ; \quad S = 0 \\ Z = \dfrac{S+1}{\sqrt{\frac{n(n-1)(2n+5)}{18}}}; & S < 0 \end{cases}, \tag{5}$$

In type: S is the normal distribution, Var(S) is variance, and its formula is:

$$\text{Var(s)} = [n(n-1)(2n+5) - \sum_{1}^{m} t_k(t_k - 1)(2t_k + 5)]/18 \tag{6}$$

In type: m is the number of groups, and $t_k$ is the number of groups with the same data. When $n$ is greater than 10, the formula of the standardized statistics is:

$$Z = \begin{cases} \dfrac{s-1}{\sqrt{\text{Var(s)}}}; & if \ \text{S} > 0 \\ 0 & ; \quad if \ \text{S} = 0 \\ \dfrac{s+1}{\sqrt{\text{Var(s)}}}; & if \ \text{S} < 0 \end{cases} \tag{7}$$

During the test, if Z is greater than 0, it indicates a significant upward trend in the series; otherwise, it indicates a significant downward trend in the series. [42]. If we set the significance levels $\alpha = 0.05$, if $|Z| \geq Z_{1-\alpha/2} = 1.96$, then the null hypothesis should be rejected, indicating that the trend of sequence change is significant; otherwise, the null hypothesis should be accepted, indicating that the trend of sequence change is not significant [43].

### 2.4. Two-Stage Annual Precipitation Partitioning

Assuming that changes in soil water storage can be ignored on an annual scale, precipitation can be divided into two components, namely runoff and evapotranspiration. The water balance can be expressed as:

$$P = Q + E, \tag{8}$$

where $P$ is precipitation, $Q$ is runoff, and $E$ is evapotranspiration.

Following the decomposition method given by L'vovich [44], long-term sedimentation can be broken down into two stages. In the first stage, precipitation is suspended as surface runoff ($Q_s$) and soil wetting ($W$), which can be expressed as:

$$P = Q_s + W, \tag{9}$$

In the second stage, the soil wetting is divided into baseflow ($Q_b$) and evapotranspiration ($ET$), which can be expressed as:

$$W = Q_b + ET, \tag{10}$$

where the sum of $Q_s$ and $Q_b$ is the total runoff ($Q$).

Based on the two-stage precipitation partitioning method and the SCS curve number method [45], Ponce and Shetty [46] proposed a method to estimate $Q_s$ and $Q_b$ at the annual scale, and Chen and Wang [47] presented an estimation method on a seasonal scale. In this paper, we focused on the annual scale, and the model can be estimated as:

$$Q_s = \frac{\left(P - \lambda_s W_p\right)^2}{P + (1 - 2\lambda_s)W_p} \; (while \; P > \lambda_s W_p), \tag{11}$$

$$Q_b = \frac{\left(W - \lambda_b V_p\right)^2}{W + (1 - 2\lambda_b)V_p} \; (while \; W > \lambda_b V_p), \tag{12}$$

where $W_p$ is the maximum of the total soil wetting, the initial wetting can be explained as a percentage ($\lambda_s$) of the soil wetting capacity ($W_p$), and $\lambda_s W_p$ can be defined as the minimum threshold for precipitation required for surface runoff. Similarly, the wetting threshold for the base flow can be defined as $\lambda_b V_p$.

### 2.5. The Nash-Sutcliffe Efficiency Coefficient

We used the Nash-Sutcliffe Efficiency coefficient ($NSE$) function [48] to evaluate model performance:

$$NSE = 1 - \frac{\sum_{i=1}^{n}\left(Q_{obs}^i - Q_{est}^i\right)^2}{\sum_{i=1}^{n}\left(Q_{obs}^i - Q_{ave}\right)^2}, \tag{13}$$

where $Q_{obs}$ is the measured runoff, $Q_{est}$ is the estimated runoff, and $Q_{ave}$ is the mean value of the measured runoff. *NSE* values can range from $-\infty$ to 1. If *NSE* = 1, it indicates that the model can perfectly model the measured data, while at a *NSE* higher than 0.5 the model performance is satisfactory [49].

### 2.6. Determining the Contribution Rate of Climate Change and Human Activities

As discussed, the datasets were divided into two periods: the pre-change period and post-change period. The pre-change period was the period prior to the point of variation, when there was no significant trend of increase or decrease in total streamflow/surface runoff/baseflow. The post-change period was the period after the point of variation, where there was a clear trend of increase or decrease in total streamflow/surface runoff/baseflow compared to the pre-change period. The proposed quantifying method assumed no human activities for the watershed in the pre-change period. In other words, during the pre-change period, if the total stream flow/surface runoff/base flow changed, it would only be attributed to climate change. During the post-change period, if the total stream flow/surface runoff/base flow changed, it could be caused by climate change or human activities, or both. Under this assumption, the model parameters were based on the observations in the pre-change period. If there were no human activities, the parameters in the post-change period were assumed to be the same as those in the pre-change period. Using the pre-change model parameters and post-change meteorological data, the total stream flow/surface runoff/base flow in the post-change period would be reconstructed, which would not be affected by human factors. The gap between the reconstructed total stream flow/surface runoff/base flow series in the post-change period and the observed ones in the pre-change period could indicate the total stream flow/surface runoff/base flow change caused by climate change. In addition, the gap between the reconstructed total stream flow/surface runoff/base flow series and the observed ones in the post-change period could indicate the total stream

flow/surface runoff/base flow change caused by human activities. The quantifying method can be computed as [50,51]:

$$\Delta Q_t = \overline{Q_o} - Q_o, \tag{14}$$

$$\Delta Q_c = \overline{Q_o} - Q_m \tag{15}$$

$$\Delta Q_h = Q_m - Q_o \tag{16}$$

$$\eta_h = \Delta Q_h / \Delta Q_t \times 100\% \tag{17}$$

$$\eta_c = \Delta Q_c / \Delta Q_t \times 100\% \tag{18}$$

where $\Delta Q_t$ represents the total change in the total stream flow/surface runoff/base flow, $\Delta Q_c$ and $\Delta Q_h$ indicate the changes in total stream flow/surface runoff/base flow caused by climate change and human activities, respectively, $\overline{Q_o}$ and $Q_o$ are the observed total stream flow/surface runoff/base flow values in the pre-change period and the post-change period, respectively, $Q_m$ is the modeled total stream flow/surface runoff/base flow values in the post-change period, and $\eta_c$ and $\eta_h$ are the climate change and impacts of human activities on total stream flow/surface runoff/base flow changes, respectively.

The advantages of the quantifying method are simple, practical, and can obtain accurate results according to the simulated values, but there is no breakdown of the impact of human activity, such as river damming, irrigation, and urban water diversion.

## 3. Results

This section provides a concise and accurate description of the experimental results, their interpretation, and some experimental conclusions that can be drawn.

### 3.1. Base Flow and Surface Runoff

The daily observed total streamflow was decomposed into surface runoff and base flow using the one-parameter digital filter method with the filter parameter value of 0.925 [37], which is widely used in the study area [40]. The observed total annual flow, surface runoff, and base flow at Linjiacun, Xianyang, and Huaxian stations across the WRB are presented in Figure 2. For the three stations, the total stream flow, surface runoff, and base flow at the annual scale tended to decline. The total stream flow decreased faster than either the surface runoff or the base flow, with the surface runoff decreasing the slowest. Both the total flow and base flow were the highest at the Xianyang station. However, its base flow index was almost the same as that at the Huaxian station, with mean annual values of 0.67 at the Xianyang station and 0.66 at the Huaxian station. The evaluation of the base flow index was similar at the Xianyang and Huaxian stations. A significant change for the base flow index at the Linjiacun station occurred in 1970. The mean annual base flow index at Huaxian and Xianyang was 0.67 with a standard deviation of 0.05, while the mean annual base flow index at Linjiacun was 0.63 with a standard deviation of 0.13 (Figure 3). The base flow index was substantially influenced by human activities, climate change, precipitation, and evaporation [40]. Therefore, it is necessary to determine the variation points of total flow, surface runoff, and base flow series for further contribution analysis.

### 3.2. Breakpoint Analysis

Breakpoints of annual total stream flow, surface runoff, and base flow were detected by the Mann-Kendall test for all three stations from upstream to downstream across the WRB, with significance levels set at $\alpha = 0.05$ ($Z_{1-\alpha/2} = 1.96$), and the results are shown in Table 1. Breakpoints of the total stream flow at the Linjiacun, Xianyang, and Huaxian stations occurred in 1990, 1986, and 1987, respectively. Breakpoints of the surface runoff and base flow at those stations were in 1986, 1989, and 1991, and 1977, 1986, and 1974, respectively. According to the results, the significant change for base flow was earlier than total stream flow, indicating that the base flow was more sensitive to the effects of climate change and human activities than the total stream flow. The breakpoint in the Huaxian station was the earliest, in 1974, among all the watersheds. This is consistent with the base

flow index changes in 1970. Therefore, we selected the 1970 points as the variation point to quantify the influence of climate change and human activities on total stream flow, surface runoff, and base flow [52].

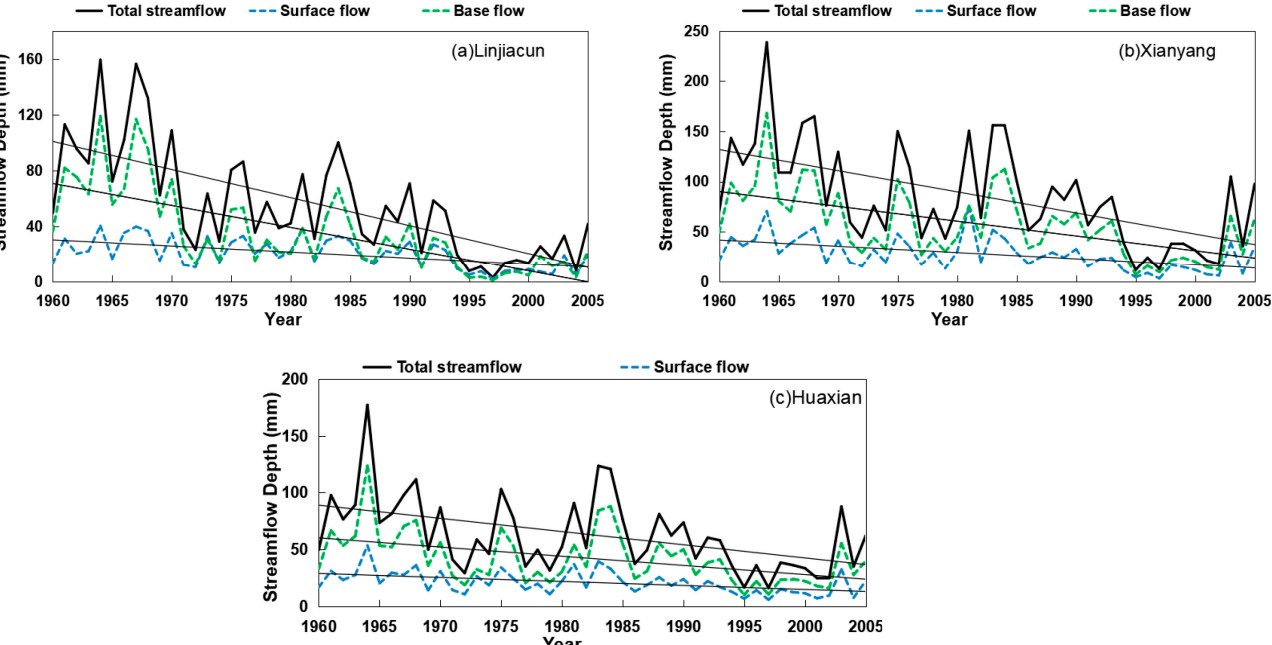

**Figure 2.** The annual total streamflow, direct runoff, and base flow in (**a–c**).

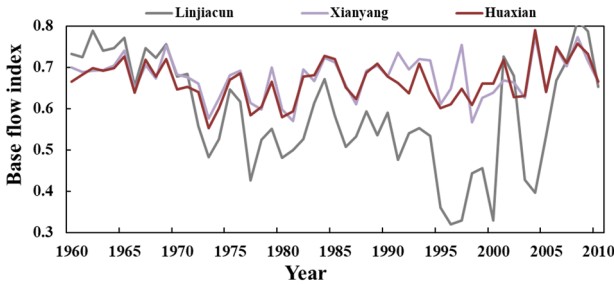

**Figure 3.** Baseflow Index Change chart of Linjiacun Station, Xianyang Station, and Huaxian Station.

**Table 1.** The breakpoint detected results of the Mann-Kendall test method.

| Station | $\alpha$ | Z | Breakpoint | | |
|---|---|---|---|---|---|
| | | | **Total Streamflow** | **Base Flow** | **Surface Runoff** |
| Linjiacun | 0.05 | ±1.96 | 1990 | 1977 | 1986 |
| Xianyang | 0.05 | ±1.96 | 1986 | 1986 | 1989 |
| Huaxian | 0.05 | ±1.96 | 1987 | 1974 | 1991 |

### 3.3. Model Calibration and Stream Flow Reconstruction

The data were divided into two periods, 1960–1970 (pre-change period) and 1971–2005 (post-change period), based on the results of the breakpoint tests to quantify the impact of climate change and humans on surface runoff and baseflow. A two-stage annual precipitation partitioning model was used to estimate the surface runoff and base flow for the post-change period. The model has four parameters: $\lambda_s$, $Wp$, $\lambda_b$, and $Vp$, which need to be estimated. We chose to estimate the parameters in an 11-year period (the whole pre-change period: 1960–1970) instead of in a six-year calibration period (1960–1965) to minimize the uncertainty in the limited data. The values of surface runoff and base flow in the natural

period from 1960 to 1970 were used to estimate the model parameters: $\lambda_s$, Wp, $\lambda_b$, and Vp. The parameters for the three stations in the WRB are listed in Table 2. As shown in Table 2, the parameters $\lambda_s$ and $\lambda_b$ do not indicate a significant pattern.

**Table 2.** Parameter evaluation of the two-stage annual precipitation partitioning model for 3 hydrological stations across the Weihe River.

| Site | Wp | Vp | $\lambda_b$ | $\lambda_s$ |
|---|---|---|---|---|
| Linjiacun | 8645 | 3000 | 0.01 | 0.01 |
| Xianyang | 8645 | 2616 | 0.01 | 0.02 |
| Huaxian | 4543 | 3983 | 0.06 | 0.02 |

The values of NSE and the coefficient of determination ($R^2$) were used to evaluate the model performance of the base flow and surface runoff. NSE and $R^2$ were computed in the validation period (1960–1970) for the three sites and are summarized in Table 3. The $R^2$ values ranged from 0.9 to 0.98, and the NSE values ranged from 0.89 to 0.98. To illustrate the model performance, we used the Linjiacun station as an example (Figure 4). The results showed that the model performed well, although the model led to an underestimation of the results observed during the peak period. Overall, the accuracies of the model calibration and validation were acceptable for the annual surface runoff and base flow estimation in the three stations.

**Table 3.** The $R^2$ and Nash-Sutcliffe Efficiency coefficient (NSE) for two-stage annual precipitation partitioning model at the 3 hydrological stations.

| Site | Base Flow (mm) | | Surface Runoff (mm) | |
|---|---|---|---|---|
| | $R^2$ | NSE | $R^2$ | NSE |
| Linjiacun | 0.90 | 0.89 | 0.96 | 0.92 |
| Xianyang | 0.96 | 0.96 | 0.97 | 0.90 |
| Huaxian | 0.98 | 0.98 | 0.91 | 0.90 |

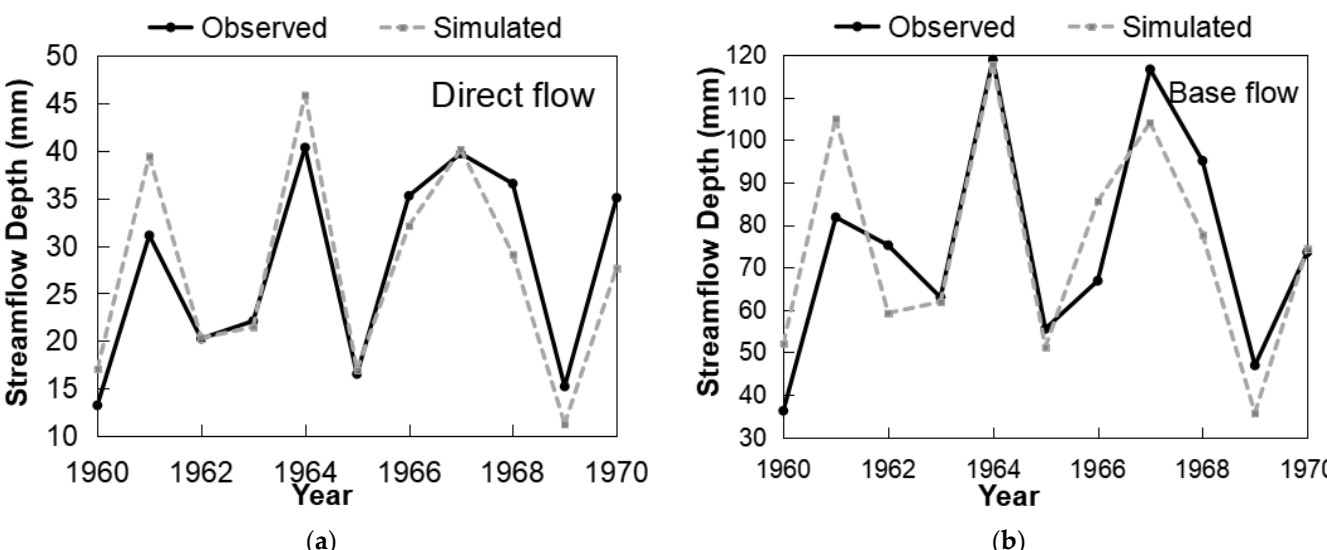

**Figure 4.** The comparison of modeled and observed surface runoff (**a**) and base flow (**b**) from 1960 to 1970 at Linjiacun.

Based on the high accuracy of the model estimates, the surface runoff and base flow in the post-change period were reconstructed using pre-change parameter estimates and post-change meteorological data from Linjiacun, Xianyang, and Huaxian stations. Figure 5 shows a comparison of the modeled and observed base flow and surface runoff during

the entire period in the WRB. During the pre-change period, data clustered around the 1:1 lines, indicating a good agreement between the modeled and observed data. However, deviations from the 1:1 line were found during the impacted period indicating human activities on base flow. Moreover, the figure shows that human activities affected the base flow more strongly than the surface runoff. The annual precipitation and observed and modeled surface runoff and base flow are shown in Figure 6. The differences between the observed and modeled base flow became larger over time, while the differences between those of the surface runoff remained small.

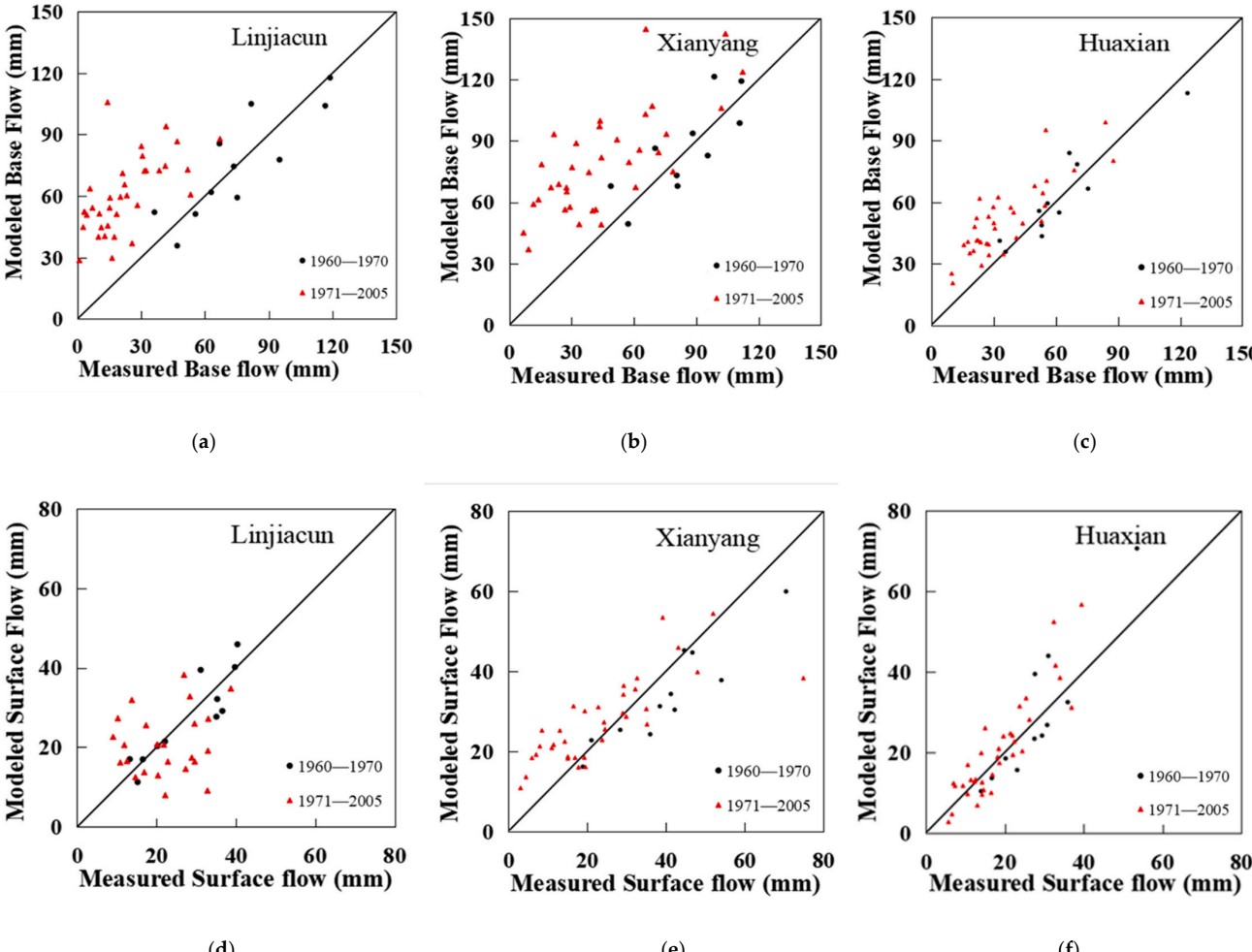

**Figure 5.** The comparison of the surface runoff and base flow were reconstructed using the parameter estimation from the pre-change period and the meteorological data for the post-change period at Linjiacun, Xianyang, and Huaxian stations across the WRB. The black lines are a 1:1 line. (**a**). measured and modeled baseflow at Linjiacun station; (**b**). measured and modeled baseflow at Xianyang station; (**c**). measured and modeled baseflow at Huaxian station; (**d**). measured and modeled surface flow at Linjiacun station; (**e**). measured and modeled surface flow at Xianyang station; (**f**). measured and modeled surface flow at Huaxian station.

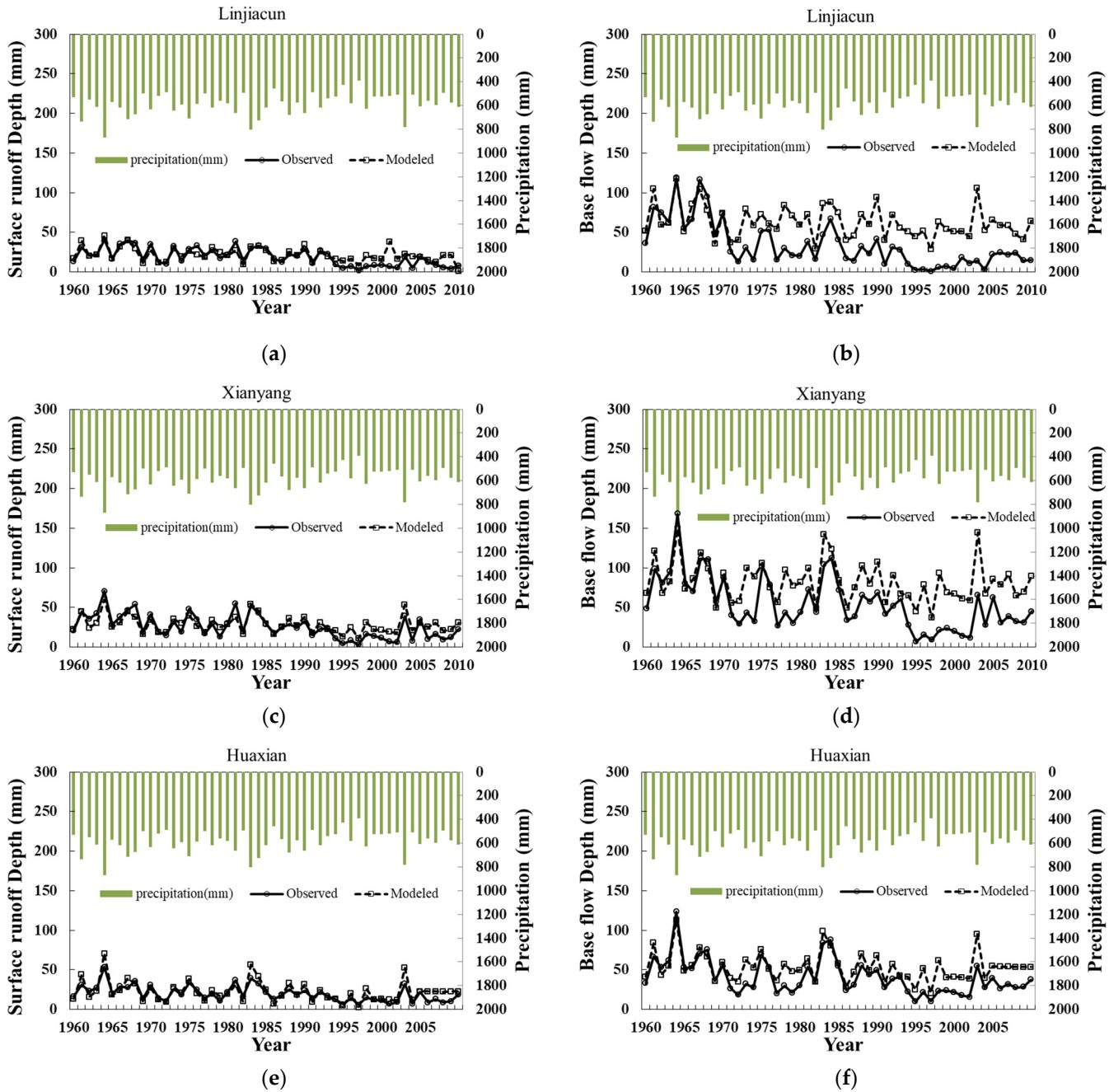

**Figure 6.** The observed and modeled annual surface runoff and base flow at Linjiacun, Xianyang, and Huaxian stations across the WRB. (**a**). observed and modeled surface flow at Linjiacun station; (**b**). observed and modeled baseflow at Linjiacun station; (**c**). observed and modeled baseflow at xianyang station; (**d**). observed and modeled baseflow at Xianyang station; (**e**). observed and modeled surface flow at Huaxian station; (**f**). observed and modeled baseflow at Huaxian station.

### 3.4. Attribution Analysis

Based on the reconstructed surface runoff and base flow data (Figure 6), the reconstructed total stream flow was generated by summing the reconstructed surface runoff and the base flow. The gap between the reconstructed total stream flow/surface runoff/base flow series in the post-change period and the observed ones in the pre-change period could indicate the change caused by climate change. The gap between the reconstructed total stream flow/surface runoff/base flow series and the observed ones in the post-change

period could indicate the change caused by human activities. Climate change and human activities have different impacts on total runoff, surface runoff, and baseflow changes.

Table 4 shows the effects of climate change and human activities on the total flow changes at the three WRB sites. As the Table 4 shown, during the post-change period, under the two effects, the total stream flow of the Linjiacun, Xianyang, and Huaxian stations decreased significantly, with the values of 61.55 mm, 62.50 mm and 35.92 mm, respectively. The two effects on total stream flow were mainly attributed to human activities at Linjiacuan and Xianyang stations, accounting for 64% and 60%, respectively, while the effect at Huaxian station was 49%. The effects of climate change and human activities on stream flow in WRB have been computed in several studies, and the results of this study are consistent with those of Zuo [32], Zhan [33], and Jiang [34]. Therefore, in the change process of total stream flow, except for the variation at Huaxian Station that was greatly affected by climate change, the other two stations were heavily influenced by human activities.

**Table 4.** Effects of climate change and human impacts on the annual total streamflow at the Linjiacun, Xianyang, and Huaxian stations.

| Site | Period | Total (mm) | Reconstructed (mm) | Total Change (mm) | Human | | Climate | |
|------|--------|-----------|--------------------|--------------------|-------|---|---------|---|
| | | | | | Values | % | Values | % |
| Linjiacun | 1960–1970 | 103.17 | 102.38 | | 39.63 | 64 | 21.92 | 36 |
| | 1971–2005 | 41.63 | 81.26 | 61.55 | | | | |
| Xianyang | 1960–1970 | 132.06 | 125.54 | | 37.59 | 60 | 24.91 | 40 |
| | 1971–2005 | 69.55 | 107.15 | 62.50 | | | | |
| Huaxian | 1960–1970 | 90.07 | 90.83 | | 17.73 | 49 | 18.20 | 51 |
| | 1971–2005 | 54.14 | 71.87 | 35.92 | | | | |

The effects of climate change and human activities on surface runoff and base flow of the three stations are summarized in Tables 5 and 6. Compared to the values of the Linjiacun, Xianyang, and Huaxian stations in the pre-change period, during the post-change period, the surface runoff decreased by 9.1 mm, 16.5 mm, and 9.6 mm, respectively, while the base flow decreased by 52.4 mm, 46.1 mm, and 26.3 mm, respectively. The surface runoff changes of the three stations are two or more times higher than the base flow change. The decrease in surface runoff in Linjiacun, Xianyang, and Huaxian was mainly affected by climate change, accounting for 76%, 77%, and 78%, respectively, while the contribution rate of human activities to base flow was 71%, 73%, and 59%, at Linjiacun, Xianyang, and Huaxian hydrological stations, respectively; so, human activities were the main driving factor of the base flow variation. Climate change controls the surface runoff change and human activities control the base flow change, and the results of this study are consistent with those of Zhang [53]. In summary, climate change greatly affects the change of surface runoff, while human activities greatly influence the change of baseflow.

**Table 5.** Effects of climate change and human impacts on the annual surface runoff at the Linjiacun, Xianyang, and Huaxian stations.

| Site | Period | Surfacel (mm) | Reconstructed (mm) | Surface Change (mm) | Human | | Climate | |
|------|--------|--------------|--------------------|---------------------|-------|---|---------|---|
| | | | | | Values | % | Values | % |
| Linjiacun | 1960–1970 | 27.75 | 27.40 | | | | | |
| | 1971–2005 | 18.62 | 20.81 | 9.13 | 2.19 | 24 | 6.94 | 76 |
| Xianyang | 1960–1970 | 40.17 | 33.79 | | | | | |
| | 1971–2005 | 23.72 | 27.49 | 16.45 | 3.77 | 23 | 12.68 | 77 |
| Huaxian | 1960–1970 | 28.13 | 28.95 | | | | | |
| | 1971–2005 | 18.54 | 20.64 | 9.59 | 2.10 | 22 | 7.49 | 78 |

**Table 6.** Effects of climate change and human impacts on the annual base flow at the Linjiacun, Xianyang, and Huaxian stations.

| Site | Period | Base (mm) | Reconstructed (mm) | Base Change (mm) | Human Values | % | Climate Values | % |
|---|---|---|---|---|---|---|---|---|
| Linjiacun | 1960–1970 | 75.42 | 74.98 | | | | | |
| | 1971–2005 | 23.02 | 60.22 | 52.40 | 37.20 | 71 | 15.20 | 29 |
| Xianyang | 1960–1970 | 91.89 | 91.75 | | | | | |
| | 1971–2005 | 45.79 | 79.44 | 46.10 | 33.65 | 73 | 12.45 | 27 |
| Huaxian | 1960–1970 | 61.94 | 28.95 | | | | | |
| | 1971–2005 | 35.64 | 51.16 | 26.30 | 15.52 | 59 | 10.78 | 41 |

## 4. Discussion

The comparison of changes in total stream flow, surface flow, base flow, and precipitation at the Linjiacun, Xianyang, and Huaxian stations in the pre-change period and the post-change period are shown in Table 7. As seen in Table 7, compared to the pre-change period, the variables in the post-change period decreased. Although the values of total stream flow, surface runoff, base flow, and precipitation decreased, the total stream flow and base flow indicated significant change, with the percentage ranging from 40% to 61% and from 43% to 70%, respectively, while the surface runoff change was slight, with the percentage ranging from 34% to 41%. The decreased values and percentage of the total stream flow and the base flow along the main stream of the WRB displayed a decreasing change from upstream to downstream, with the highest value at Linjiacun station and the lowest one at Huaxian station. Compared to the flow change, the precipitation change was a small range, from 9% to 11%. The surface runoff change was mainly controlled by climate change.

**Table 7.** The comparison of changes for total stream flow, surface flow, base flow, and precipitation at the Linjiacun, Xianyang, and Huaxian stations in the pre-change period and post-change period.

| Index | Linjiacun | | Xianyang | | Huaxian | |
|---|---|---|---|---|---|---|
| | Change (mm) | Percentage (%) | Change (mm) | Percentage (%) | Change (mm) | Percentage (%) |
| Total streamflow | 63.73 | 61 | 62.50 | 47 | 35.92 | 40 |
| Surface flow | 9.85 | 35 | 16.45 | 41 | 9.59 | 34 |
| Base flow | 52.88 | 70 | 46.05 | 50 | 26.34 | 43 |
| Precipitation | 63.91 | 11 | 55.98 | 9 | 58.94 | 9 |

In addition, a large number of human activities have been implemented since 1970. Soil conservation practices in 1970 covered 2500 km$^2$; however, the area in 2006 was 33,344 km$^2$, increased by 30,844 km$^2$ [34]. It was almost 13 times as large as that in 1970. From 1970 to 2005, just in the Shaanxi province, seven hyper-irrigation areas began irrigation, specifically Baojixia and Jinghuiqu in 1972, Fengjiashan in 1974, Shibaochuan in 1975, Yangmaowan in 1978, Taoqupo in 1980, and Shitouhe in 1981 with a total area of approximately 5900 km$^2$. Since 1994, the population of the Shaanxi Province has increased by 3.02 million from 20.24 million to 23.26 million, almost 13%. Because of the shortage of water resources, the government has invested human and financial resources in the South-North Water Transfer Project, or WRB, from the Han River to Shaanxi Province [54]. In the Gansu province, rainwater-harvesting agriculture has made great progress, and a number of water harvesting engineering projects started in the 1980s [55]. Su et al. [56] reported rainwater collection, water and soil conservation, and reservoir management are the main factors effecting the stream flow change in the WRB. All the mentioned human activities can cause the stream flow to change directly or indirectly through affecting land use and land cover.

## 5. Conclusions

In this study, we quantified the effects of climate change and human activities on the total streamflow, surface runoff, and base flow from 1960 to 2005 measured at three hydrological stations: Linjiacun, Xianyang, and Huaxian station in the WRB. Using the breakpoint, the annual data were divided into two periods: the pre-change period and the post-change period. The two-stage model was calibrated and verified by using the pre-change period data, and watershed hydrology was reconstructed for the post-change period. The differences in our study indicated that the main driving factor for the decrease in surface runoff was climate change, contributing over 76%. The contribution rate of human activities to base flow was 71%, 73%, and 59%, respectively; so, human activities were the main driving factor of base flow variation. Overall, climate change and human impact have different effects on different components of the stream flow. For surface flow, climate change has greater impacts than humans; however, for base flow, human activities have greater impacts than climate change. Watershed management in the WRB must incorporate appropriate measures to mitigate the likely impacts of climate change on watershed hydrology and achieve the desired goals of ecological restoration.

This study focuses on the change of historical flow of the Weihe River Basin but does not carry out corresponding research into and prediction of future flow changes. Therefore, future research will focus on the prediction of the future flow of the river basin.

**Author Contributions:** Conceptualization, Z.M. and B.Z.; methodology, J.F.; software, Y.L.; validation, T.Q. and Y.C.; formal analysis, S.L.; investigation, resources, J.F.; data curation, G.L.; supervision, Y.C.; project administration, T.Q. All authors have read and agreed to the published version of the manuscript.

**Funding:** This research was supported by the National Key Research and Development Project (Grant No. 2017YFA0605004) , the Innovative Research Group of Heibei Natural Science Foundation, (Grant No. E2019402432, E2019402102), National Science Fund for Young Scholars (Grant No. 52009053), the National Natural Science Foundation of China (Grant No. U1802241), the open project of State Key Laboratory Base of Eco-Hydraulic Engineering in Arid Area, ( Grant No. 2019KFKT-4), and Hebei Key Laboratory of Intelligent Water Conservancy in Hebei University of Engineering in Handan.

**Conflicts of Interest:** The authors declare no conflict of interest.

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
