# Peer review of "Base Flow Variation and Attribution Analysis Based on the Budyko Theory in the Weihe River Basin"

_water, doi:10.3390/w14030334_

Round 1

Reviewer 1 Report

The primary focus of the artilce is the estimation of the human and climate effects on base flow and surface runoff in the Weihe River basing. The title does not reflect the article content properly therefore should be change to something like this: „the estimation of the human and climate effects on base flow and surface runoff in the Weihe River basing using Budyko theory”. The article brings some new scientific findings regarding the modelling of surface runoff and base flow. Link to Water is clear in Title, Abstract, Introduction, and Results. The article contains a short discussion and really compares the results with international findings.  The abstract represents the article content properly. The aim and methodology is very clearly specified so as the main results. The topic fits in the scope of the journal’s scope. The introduction is wisely connect climate and human issues with water management. The research gap is also stated clearly. The theoretical part of the article is all rigth, it contains 55 sources (mainly from databases such as Scopus and Web of Science). 84% of the sources are not older than 15 years. Materials and Methods section contains a lot of mistakes that should be corrected. These are given in the corrected PDF file. The figures and tables are usful for the readers and of acceptable quality. There are some mistakes (grammar and typing, etc…) which can be seen from the corrected pdf version which I will provide for the authors. It is especially annoying to see many „hypen” mistakes in the document. With repsect to the methodology, the breakpoint analysis was not described in a clear way and should be explained in more details. Results are clearly formulated and supported by appropriate tables or pictures. There were no scientific hypotheses formed and evaluated. Limitation of the study should also be mentioned in the conclusion. All in all, the article is written in an interesting way and satisfies the formal requirements. I think the refecences and the template is according to the journal requirements but some references should be corrected and I don’t understand why the authors denoted the year of some articles with bold. The article is written in good english and minor spellcheck and proofreading is required by a native english speaker. All the mistakes were indicated in the PDF file and will be available for the authors. I see a good potentials in this article and I would suggest a minor revision before publishing.

Minor issues:

in line 18 there are double points

In line 98 (m3) use subscript in case of 3

In line 105 correct four hydrogeological stations to three stations

Hypen mistake: in lines 111,113,131-133,137 and elswhere in the document

In line 117 you mention the annual aggregation, why was it annual?

In line 131 change „in type” to „where” and T-1 is t-1.

In line 137 use lower subscript

In formula 3 the second (inner) sum is not correct, I mean the indice should go from j=1 to i-1. Perhaps this is the reason the outer sum is going from i=2.. You also have to explain sgn.

In formula 4 capital S should be used.

Formula 5 is incorrect in many ways: the sum contains „m” which should be „k”. There are two different type of parentheses without a pair.

In formula 6: capital S should be used

In lines 145-147 there are multiple errors: significance level should be denoted by alpha and use lower subscript. The wording is not correct, use „the null hypothesis should be rejected” or „accepted”.

In line 170 use lower subscript for obs, ave, est

In formula 14 Qm is not defined.

In line 201: The order is not good (human comes first and then the climate change, so keep this order)

A more detailed explanation is needed how did you perform this analysis. For example you split the whole interval into two parts and tested the trend with Mann-Kendall test and compared the Z value. Then you moved the split point till the end of the interval and test again. What was your criteria for detecting the break point in terms of Z values for pre and post-change periods?

In table 1, Z score and p-value should be reported

Space is required between the value and „mm” every where for example in lines 303,316

References should be checked, for example see line 482

Author Response

Responses to comments on Paper water-1546607

Title: Base flow variation and attribution analysis based on Budyko theory

Authors: Zheng Mu, Guanpeng Liu, Shuai Lin, Jingjing Fan *, Tianling Qin *,

Yunyun Li, Yao Cheng, Bin Zhou

Manuscript ID: water-1546607 - Revision Reminder

 We greatly appreciate the efforts put in by you and Reviewers and thank you all for the words of encouragement, constructive comments and insightful suggestions that greatly enrich the quality of our manuscript. In this revision, the point-by-point responses to Reviewers’ and Associate Editor‘s comments and suggestions are described below, and the relevant corrections are incorporated into the revision of the manuscript accordingly (colored in red). We hope that the revision of the manuscript and our accompanying responses will be sufficient to make our manuscript suitable for publication in your esteemed Water.

Here are our responses one-by-one.

Response to the Reviewer #1:

The primary focus of the artilce is the estimation of the human and climate effects on base flow and surface runoff in the Weihe River basing. The title does not reflect the article content properly therefore should be change to something like this: „the estimation of the human and climate effects on base flow and surface runoff in the Weihe River basing using Budyko theory”. The article brings some new scientific findings regarding the modelling of surface runoff and base flow. Link to Water is clear in Title, Abstract, Introduction, and Results. The article contains a short discussion and really compares the results with international findings.  The abstract represents the article content properly. The aim and methodology is very clearly specified so as the main results. The topic fits in the scope of the journal’s scope. The introduction is wisely connect climate and human issues with water management. The research gap is also stated clearly. The theoretical part of the article is all rigth, it contains 55 sources (mainly from databases such as Scopus and Web of Science). 84% of the sources are not older than 15 years. Materials and Methods section contains a lot of mistakes that should be corrected. These are given in the corrected PDF file. The figures and tables are usful for the readers and of acceptable quality. There are some mistakes (grammar and typing, etc…) which can be seen from the corrected pdf version which I will provide for the authors. It is especially annoying to see many „hypen” mistakes in the document. With repsect to the methodology, the breakpoint analysis was not described in a clear way and should be explained in more details. Results are clearly formulated and supported by appropriate tables or pictures. There were no scientific hypotheses formed and evaluated. Limitation of the study should also be mentioned in the conclusion. All in all, the article is written in an interesting way and satisfies the formal requirements. I think the refecences and the template is according to the journal requirements but some references should be corrected and I don’t understand why the authors denoted the year of some articles with bold. The article is written in good english and minor spellcheck and proofreading is required by a native english speaker. All the mistakes were indicated in the PDF file and will be available for the authors. I see a good potentials in this article and I would suggest a minor revision before publishing.

Major Comments

1in line 18 there are double points.

ResponseThe extra dot on line 18 has been removed.

2In line 98 (m3) use subscript in case of 3

Response: Treating 3 as a superscript, the result is m3

3In line 105 correct four hydrogeological stations to three stations

Response: Four has been changed to three.

4Hypen mistake: in lines 111,113,131-133,137 and elswhere in the document

Response: All hyphenation errors are corrected.

5In line 117 you mention the annual aggregation, why was it annual?

Response: In this study, we focus on the annual scale, we will continue on month scale on the next work.

6In line 131 change „in type” to „where” and T-1 is t-1

Response: Changed in-type and T-1 to where and t-1 respectively

7In line 137 use lower subscript

Response: Modified into subscript labels, and renew all the former.

8In formula 3 the second (inner) sum is not correct, I mean the indice should go from j=1 to i-1. Perhaps this is the reason the outer sum is going from i=2.. You also have to explain sgn.

Response: The formula has been corrected and the definition of sgn has also been added,the result is

9In formula 4 capital S should be used.

Response: s has been changed to S.

10Formula 5 is incorrect in many ways: the sum contains „m” which should be „k”. There are two different type of parentheses without a pair.

Response: The formula has been corrected and corrected the definition of the formula. the result is

Where m is the number of groups and tk is the numb-er of groups with the same data.

11In formula 6: capital S should be used

Response: s has been changed to S.

12In lines 145-147 there are multiple errors: significance level should be denoted by alpha and use lower subscript. The wording is not correct, use „the null hypothesis should be rejected” or „accepted”.

Response: The question of α and subscripts has been revised, and the following content has been re-described: Set significance levels α=0.05, if |Z|≥Z1-α/2=1.96, then the null hypothesis should be rejected, indicating that the trend of sequence change is significant; otherwise, the null hypothesis should be accepted, indicating that the trend of sequence change is not significant.

13In line 170 use lower subscript for obs, ave, est

Response: The question of subscripts has been revised

14In formula 14 Qm is not defined.

Response: Supplementary definition of Qm: Qm is the modeled total stream flow / surface runoff / base flow values in the post-change period

15In line 201: The order is not good (human comes first and then the climate change, so keep this order)

Response: Arranged the order to be the same as before

16A more detailed explanation is needed how did you perform this analysis. For example you split the whole interval into two parts and tested the trend with Mann-Kendall test and compared the Z value. Then you moved the split point till the end of the interval and test again. What was your criteria for detecting the break point in terms of Z values for pre and post-change periods?

Response: According to this question, we add Z value on the paragraph. The Mann-kendall test in this study is to detect the variation point, and α=0.05, Z1-α/2=±1.96, if the intersection appear in the gap of the ±1.96, the intersection is the variation point, otherwise, we set the intersection is not the real one.

17In table 1, Z score and p-value should be reported

Response: Added Z score in the table. For p_value is usually used in correlation analysis of two variables, so I did not add this variable in the table.

18Space is required between the value and „mm” every where for example in lines 303,316

Response: Added space between the value and ‘mm’

19References should be checked, for example see line 482

Response: References have been checked and corrected for errors

Reviewer 2 Report

The paper deals with an internationally relevant topic by using appropriate analystical methods, with a comprehensive result presentation and sound conclusions. The structure is logical, the text easy to follow, the abstract informative. The illustrations are fine, the tables and the references are relevant. The preparation of the manuscript is meticulous.

The text needs a careful control for typos (some I marked in the attached pdf). The English is understandable, but requires minor corrections that can be fixed by a native English speaker.

Some corrections, comments and suggestions are marked in the attached pdf. file.

Overall, I recommend the study for publication in Water. A professional work done.

Author Response

Responses to comments on Paper water-1546607

Title: Base flow variation and attribution analysis based on Budyko theory

Authors: Zheng Mu, Guanpeng Liu, Shuai Lin, Jingjing Fan *, Tianling Qin *,

Yunyun Li, Yao Cheng, Bin Zhou

Manuscript ID: water-1546607 - Revision Reminder

 We greatly appreciate the efforts put in by you and Reviewers and thank you all for the words of encouragement, constructive comments and insightful suggestions that greatly enrich the quality of our manuscript. In this revision, the point-by-point responses to Reviewers’ and Associate Editor‘s comments and suggestions are described below, and the relevant corrections are incorporated into the revision of the manuscript accordingly (colored in red). We hope that the revision of the manuscript and our accompanying responses will be sufficient to make our manuscript suitable for publication in your esteemed Water.

Here are our responses one-by-one.

Response to the Reviewer #2:

The paper deals with an internationally relevant topic by using appropriate analystical methods, with a comprehensive result presentation and sound conclusions. The structure is logical, the text easy to follow, the abstract informative. The illustrations are fine, the tables and the references are relevant. The preparation of the manuscript is meticulous.

The text needs a careful control for typos (some I marked in the attached pdf). The English is understandable, but requires minor corrections that can be fixed by a native English speaker.

Some corrections, comments and suggestions are marked in the attached pdf. file.

Overall, I recommend the study for publication in Water. A professional work done.

Major Comments

1I suggest to complete the title as your study is actualistic based on the specific hydro-geographical data

ResponseThe title has been changed to ‘Base flow variation and attribution analysis based on Budyko theory in Weihe River Basin’

2Hypen mistake: in lines 51,366, and elsewhere in the document

Response: All hyphenation errors are corrected, including lines 51, 366, 367, 368, 369, 373, 374, 375, 376, 380 and 381.

3Some writing mistakes: in lines 23,47,55 ,and elsewhere in the document

Response: All writing errors are corrected, including lines 23, 47, 55, 59, 65, 76, 80, 89, 207, 208, 294, 333, 334, 335, 339, 340 and 365.

4are there some analogies of your findings (in China or elsewhere in Asia in the areas with comparable environmental conditions and human landscape use?) If so, pls add a few example with reference.

Response: added reference in the article about this result.

This manuscript is a resubmission of an earlier submission. The following is a list of the peer review reports and author responses from that submission.